# Studying the Long-term Impact of COVID-19 in Kids (SLICK). Healthcare use and costs in children and young people following community-acquired SARS-CoV-2 infection: protocol for an observational study using linked primary and secondary routinely collected healthcare data from England, Scotland and Wales

Olivia V Swann ,[1,2] Nazir I Lone,[3] Ewen M Harrison ,[1] Laurie A Tomlinson ,[4] Alex J Walker ,[5] Michael J Seaborne ,[6] Louisa Pollock,[7] James Farrell,[1] Peter S Hall,[8] Sohan Seth,[9] Thomas C Williams,[2] Jennifer Preston,[10] J. Samantha Ainsworth,[10] Freya F Semple,[11] J Kenneth Baillie,[12] Srinivasa V Katikireddi ,[13] Ashley Akbari ,[14] Ronan Lyons ,[14] Colin R Simpson ,[3,15] Malcolm G Semple,[16,17] Ben Goldacre ,[5] Sinead Brophy ,[18] Aziz Sheikh,[3] Annemarie B Docherty[1]

For numbered affiliations see end of article.

**Correspondence to**
Dr Olivia V Swann;
Olivia.Swann@ed.ac.uk

## ABSTRACT

**Introduction** SARS-CoV-2 infection rarely causes hospitalisation in children and young people (CYP), but mild or asymptomatic infections are common. Persistent symptoms following infection have been reported in CYP but subsequent healthcare use is unclear. We aim to describe healthcare use in CYP following community-acquired SARS-CoV-2 infection and identify those at risk of ongoing healthcare needs.

**Methods and analysis** We will use anonymised individual-level, population-scale national data linking demographics, comorbidities, primary and secondary care use and mortality between 1 January 2019 and 1 May 2022. SARS-CoV-2 test data will be linked from 1 January 2020 to 1 May 2022. Analyses will use Trusted Research Environments: OpenSAFELY in England, Secure Anonymised Information Linkage (SAIL) Databank in Wales and Early Pandemic Evaluation and Enhanced Surveillance of COVID-19 in Scotland (EAVE-II). CYP aged ≥4 and <18 years who underwent SARS-CoV-2 reverse transcription PCR (RT-PCR) testing between 1 January 2020 and 1 May 2021 and those untested CYP will be examined. The primary outcome measure is cumulative healthcare cost over 12 months following SARS-CoV-2 testing, stratified into primary or secondary care, and physical or mental healthcare. We will estimate the burden of healthcare use attributable to SARS-CoV-2 infections in the 12 months after testing using a matched cohort study of RT-PCR positive, negative or untested CYP matched on testing date, with adjustment for confounders. We will identify factors associated with higher healthcare needs in the 12 months following SARS-CoV-2 infection using an unmatched cohort of RT-PCR positive CYP. Multivariable

## STRENGTHS AND LIMITATIONS OF THIS STUDY

⇒ Objective, direct examination of clinician-recorded healthcare use by children and young people (CYP) post SARS-CoV-2 infection.

⇒ Population-wide coverage of all CYP <18 years in Scotland and Wales and approximately 4.8 million CYP in England.

⇒ Lack of access to SARS-CoV-2 lateral flow testing (rapid antigen testing) results may result in misattribution of SARS-CoV-2 status in patients when reverse transcription PCR testing was not performed.

⇒ Access to health services is presumed to be available for anyone who needed it, but this may have been reduced by local healthcare policies and patient health-seeking behaviour at different points during the pandemic.

⇒ Owing to the time needed for 12 months of follow-up, this study will focus on healthcare use after infection with wild-type and Alpha variants of SARS-CoV-2, which may differ from Delta and Omicron.

logistic regression and machine learning approaches will identify risk factors for high healthcare use and characterise patterns of healthcare use post infection.

**Ethics and dissemination** This study was approved by the South-Central Oxford C Health Research Authority Ethics Committee (13/SC/0149). Findings will be preprinted and published in peer-reviewed journals. Analysis code and code lists will be available through public GitHub repositories and OpenCodelists with meta-data via HDR-UK Innovation Gateway.

## INTRODUCTION

SARS-CoV-2 causes the disease COVID-19, with adults being more severely affected than children throughout the pandemic.[1] While hospitalisation with SARS-CoV-2 is rare in children and young people (CYP),[2] infection is common, with up to 70% (95% CI 68% to 71%) of 5–14 years old estimated to have been infected with SARS-CoV-2 in the UK by December 2021.[3] While research on COVID-19 in CYP has focused on index hospitalisations and deaths, this acute view means we have not established what the additional healthcare needs are for the majority of CYP after mild or asymptomatic SARS-CoV-2 infection. There is also little information on the changes in healthcare use for children with comorbidities who may be at risk of exacerbations (eg, asthma). The large numbers of CYP infected with SARS-CoV-2 in the UK means that even a small increase in healthcare use in this population could substantially impact on healthcare services. Being asymptomatic with initial infection does not guarantee against developing subsequent illness from SARS-CoV-2, for example, CYP who are asymptomatic with their initial SARS-CoV-2 infection can develop Multisystem Inflammatory Syndrome in Children 2–8 weeks later.[4] While this complication is extremely rare (approximately 3 cases per 10 000 infections),[5] it underlines the need to include CYP who are initially asymptomatic from SARS-CoV-2 infection when examining subsequent healthcare use.

A wide variety of persistent symptoms have been reported in CYP following SARS-CoV-2 infection, with studies varying in design and quality.[6] Most reports have used a questionnaire or clinic-based approach to symptom reporting, often after hospitalisation with COVID-19 or in patients self-identifying as having Long-COVID, introducing significant potential sources of bias.

Data on long-term healthcare use following SARS-CoV-2 infection is beginning to emerge, although most studies have focused on adults rather than CYP. One large study of American adults (n=5 064 270) reported an increase in outpatient healthcare use in the 6 months following SARS-CoV-2 infection (HR of 1.20 (1.19–1.21).[7] Another American study (n=250 514) found COVID-19 diagnosis was associated with an additional 0.7269 (95% CI 0.7088 to 0.7449) monthly healthcare visits (combined inpatient and outpatient visits excluding respiratory healthcare contacts) in the 6 months after diagnosis.[8] This study did include some CYP (n not given) and reported that healthcare use post-COVID-19 diagnosis increased slightly from 2–5 months after diagnosis for those ≤17 years old, but returned to prediagnosis baseline levels by 6 months.

One Norwegian study examined healthcare use in CYP aged 1–19 years (n=706 885) for 6 months from SARS-CoV-2 testing and reported an increase in primary healthcare use for all ages during the first 1–4 weeks following a positive test compared with CYP who tested negative.[9] These presentations were predominantly respiratory. This increase in healthcare use was more sustained in younger CYP, while those aged 16–19 years retuned to baseline healthcare use by 5–8 weeks. The study did not find any increase in use of specialist care for any age group.

No studies have yet examined healthcare use in CYP in the UK following SARS-CoV-2. Using routinely collected anonymised electronic health record (EHR) data at an individual level and population-scale matched by SARS-CoV-2 reverse transcription-PCR (RT-PCR) status to examine healthcare use after SARS-CoV-2 infection in CYP offers an alternative method to questionnaire or clinic-based symptom reporting after SARS-CoV-2. In addition to traditional epidemiological approaches, machine learning methods are also proving increasingly important in the analysis of large routinely collected healthcare datasets in SARS-CoV-2.[10] Using machine learning to identify clusters of patients with similar healthcare trajectories provides a complementary approach to traditional epidemiology to identify patients at risk of high healthcare use post infection. A combination of approaches would establish the long-term healthcare use attributable to SARS-CoV-2 in CYP, which is essential both for tailoring individual care for patients at risk of high healthcare use post infection and informing health service and vaccination planning.

### Aims

We aim to establish the patterns and burden of healthcare use in CYP attributable to community-acquired SARS-CoV-2 infection and identify those CYP at risk of high or ongoing healthcare needs in England, Scotland and Wales.

### Objectives

We will:
1. Describe the background healthcare use in CYP before and during the pandemic.
2. Compare healthcare use in CYP in the 12 months after testing positive, negative or not being tested for SARS-CoV-2 by RT-PCR to estimate burden of healthcare use attributable to SARS-CoV-2.
3. Identify factors associated with higher healthcare use (including having comorbidities) in the 12 months following SARS-CoV-2 infection.

## METHODS

### Study period

The period covered by the study will span 1 January 2019 to 1 May 2022 and focus on SARS-CoV-2 infections until 1 May 2021. This study period was chosen to provide 12

months of follow-up data for CYP infected to the end of the second wave of SARS-CoV-2 in the UK (end of April 2021)[11] as well as those testing negative or not tested. Inclusion of the period from 1 January 2019 to 1 January 2020 will also provide at least a year of data on prepandemic data on healthcare use for each CYP.

## Study design

The study will comprise three main approaches: a descriptive graphical analysis addressing Objective 1 (background healthcare use before and during the pandemic), a matched cohort study addressing Objective 2 (estimating healthcare use post SARS-CoV-2 infection) and an unmatched cohort study addressing Objective 3 (identifying factors associated with higher healthcare needs post SARS-CoV-2 infection).

## Study population

The study population will vary with objective:

### Inclusion criteria (all objectives)

Registered with a general practitioner (GP) in Scotland or Wales (includes all general practices) or England (The Phoenix Partnership (TPP) a group of GP practices with a unified electronic patient-record system covering approximately 34% of practices in England.)[12]

### Exclusion criteria (all objectives)

► Positive index SARS-CoV-2 RT-PCR test performed after 7 days in hospital (to exclude nosocomial infections).[13]
► CYP with discrepant SARS-CoV-2 RT-PCR results on the same date.

### Objective 1: Objective-specific inclusion criteria

► Age ≥4 years and <18 years on 1 January 2019 (prepandemic period).
► Age ≥4 years and <18 years on 1 January 2020 (pandemic period–pandemic year 1).
► Age ≥4 years and <18 years on 1 January 2021 (pandemic period–pandemic year 2).

### Objective 2: Objective-specific inclusion criteria

► Underwent SARS-CoV-2 PCR testing (or untested but matched to CYP who had been tested) between 1 January 2020 and 1 May 2021.
► Age ≥4 and <18 years on date of testing/matching.
► At least 12 months of healthcare data available both before and after SARS-CoV-2 PCR test/date of matching if not tested.
► No previous positive SARS-CoV-2 PCR test recorded.

### Objective 3 - objective-specific inclusion criteria

► Positive SARS-CoV-2 RT-PCR test between 1 January 2020 and 1 May 2021.
► Age ≥4 and <18 years on date of testing.
► At least 12 months of healthcare data available both before and after SARS-CoV-2 PCR test.
► No previous positive SARS-CoV-2 PCR test recorded.

## Data sources and validation

Data will be held securely and analyses conducted within nation-specific Trusted Research Environments (TREs): OpenSAFELY in England,[14] Secure Anonymised Information Linkage (SAIL Databank[15]) in Wales and the Early Pandemic Evaluation and Enhanced Surveillance of COVID-19 (EAVE-II) platform[16] within Public Health Scotland in Scotland. TREs provide secure computing environments which hold data remotely and enable access for analysis without the data itself ever leaving the secure site.

OpenSAFELY is a secure, transparent, open-source software platform for analysis of EHR data allowing detailed analysis of pseudonymised primary care patient records in England. Other datasets are linked within the same environment using a matching pseudonym derived from the National Health Service (NHS) number.

SAIL Databank brings together electronically held, person-based, routinely collected demographic and clinical data across Wales for the purpose of conducting and supporting health-related research, which are pseudonymised using Anonymous Linking Fields.

EAVE-II is a national-linked dataset of patient-level primary care data, out-of-hours (OOH), hospitalisation, mortality and laboratory data across Scotland. Data are held securely and analysed in the Public Health Scotland TRE.

Deterministic and probabilistic linking of datasets will be carried out via Community Health Index (CHI) number in Scotland and by NHS number in England and Wales. NHS and CHI numbers are unique identifiers used in all healthcare contacts across the NHS.[17] Datasets contributing to each country's final database are described in online supplemental table 1 with data flow diagrams in online supplemental figures A–C. In addition to the study period outlined, data from birth will also be examined to identify comorbidities, including common chronic childhood conditions.[18] In the event of missing data, these will be supplemented by information for that CYP in linked datasets. All variables will be checked for patterns of missingness and implausible values and a log maintained for reasons where records are excluded from analysis. In cases where an analysis variable has high levels of missingness, alternative variables which are closely related may be considered as a proxy for these missing data. Depending on the cause ascertained for missing variables, we will consider imputation.

As paediatric multisystem inflammatory syndrome temporally associated (PIMS-TS) is a new disease, International Classification of Diseases 10th Revision coding was not introduced until November 2020. Admission will be considered due to PIMS-TS if occurring between 1 February 2020 and 1 November 2020 and coded as Kawasaki disease, toxic shock syndrome or systemic inflammatory response (proxies for PIMS-TS) or if admitted after 1 November 2020 and coded as PIMS-TS.[19] National PIMS-TS databases (available in Scotland and Wales) will be used for sensitivity analyses. The major data sources

**Table 1** Groupings of variables by source

| | Variable | Data source England (OpenSAFELY) | Scotland (EAVE-II) | Wales (SAIL) |
|---|---|---|---|---|
| Demographics | Sex | TPP | EAVE-II | WDSD/WLGP |
| | Age | TPP | EAVE-II | WDSD/WLGP |
| | Ethnicity | TPP | EAVE-II | CENW/NCCH |
| Socioeconomic | IMD | TPP | EAVE-II | WDSD |
| Place of residence | Health board/STP, urban rural index | TPP | EAVE-II | WDSD |
| Accommodation type | Private or social housing | NA | EAVE-II | CENW |
| Comorbidities | Chronic childhood conditions | TPP, SUS APCS, SUS OPA, ISARIC | EAVE-II, SMR00/01/04/06, ISARIC | CYFI, BREC, WCSU, WLGP, PEDW |
| | SARS-CoV-2 shielding list | TPP | EAVE-II | CVSP |
| SARS-CoV-2 vaccination | Vaccine (type, date) | TPP | TVMT | CVVD |
| Laboratory tests | RT-PCR SARS-CoV-2 test (date and result) | SGSS | COVID-testing | PATD |
| | Viral variant | SGSS | COG UK | CVSD/PATD |
| Secondary care | ED contact | SUS ECDS | A+E Datamart | EDDD/EDDS |
| | Outpatient clinic contact | SUS OPS | SMR00 | OPDW |
| | Hospital admission | SUS APCS | SMR01/04 | PEDW |
| | Admission ICD-10 code | SUS APCS | SMR | PEDW |
| | Level of care | SUS/ISARIC | SMR/ISARIC | PEDW/CCDS |
| | Length of stay | SUS/ISARIC | SMR/ISARIC | PEDW/CCDS |
| | PIMS-TS | SUS/ISARIC | SMR/ISARIC | PEDW/CCDS |
| Primary care | In-hours contact | TPP | EAVE-II | WLGP |
| | Community prescriptions | TPP | EAVE-II/PIS | WDDS |
| Unscheduled care | NHS 111 contact | NA | NHS 24 | NHSO |
| | Ambulance contact | NA | SAS | WASD/NHSO |
| | GP out of hours (OOH) contact | NA | GP OOH | NHSO |
| Mortality | Death (all cause, COVID-19 main cause or <28 days of positive SARS-CoV-2 RT-PCR) | ONS deaths | NRS deaths | ONS deaths/ADDE |
| Symptoms | Presenting symptoms in CYP admitted with SARS-CoV-2 | ISARIC (subset only) | ISARIC (subset only) | ISARIC (subset only) |

ADDE, Annual District Death Extract; APCS, admitted patient care statistics; CCDS, Critical Care Data Source; CENW, Office of National Statistics Census; COG-UK, Centre of Genomics UK; CVSP, COVID-19 Shielded People; CVVD, COVID-19 Vaccine Data; EAVE-II, Early Pandemic Evaluation and Enhanced Surveillance of COVID-19; ECDS, emergency care datasets; ED, emergency department; EDDD, emergency department dataset daily; EDDS, Emergency Department Dataset; GP, general practitioner; ICD-10, International Classification of Diseases 10th Revision; IMD, Index of Multiple Deprivation; ISARIC, International Severe Acute Respiratory and emerging Infection Consortium; NA, not available; NCCH, National Community Child Health; NHS, National Health Service; NHSO, NHS 111 Call data; NRS, National Records of Scotland; ONS, Office for National Statistics; OPA, outpatient attendances; OPDW, Outpatient Dataset for Wales; PATD, Pathology Data (COVID-19 daily); PEDW, Patient Episode Database for Wales; PIMS-TS, paediatric multisystem inflammatory syndrome temporally associated ; PIS, Prescribing Information System; SAIL, Secure Anonymised Information Linkage; SAS, Scottish Ambulance Service; SGSS, Second Generation Surveillance System; SMR, Scottish Morbidity Record; STP, Sustainability and Transformation Partnership; SUS, Secondary Use Services; TPP, The Phoenix Partnership (GP group); WASD, Welsh Ambulance Service Dataset; WCSU, Welsh Cancer Incidence Surveillance Unit; WDDS, Welsh Dispensing Dataset; WDSD, Welsh Demographic Service Dataset; WLGP, Welsh Longitudinal General Practice.

for each variable are detailed in table 1 (adapted from Adeloye et al[20]).

## Exposure

The exposure of interest is diagnosis of SARS-CoV-2 infection, defined as a positive RT-PCR test result. The date of exposure is defined as the date of the positive RT-PCR test result.

## Outcomes

The primary outcome measure will be cumulative NHS healthcare costs over the 12 months following SARS-CoV-2 testing. This will provide an overarching measure that is reflective of healthcare resource use, which is expressed on a monetary scale that is common between the three nations and common to all types of activity. Activity will

only contribute to the primary outcome measure if it is quantifiable from data in all three nations. Healthcare costs will be broken down into budget-holder perspectives; secondary care (critical care/inpatient/outpatient/A&E) and primary care (face to face or telephone in-hours primary care activity). A sensitivity analysis of unscheduled care (eg, NHS 24, ambulance, GP OOH) will be undertaken for the nations where this data is available (Scotland and Wales). To ensure comparability, unit costs will be assigned from a common country (England) using Personal Social Services Research Unit costs with a common base year.[21]

Secondary outcomes will constitute units of healthcare activity, quantifiable as counts over time or rates, which can be quantified to a common definition between the three nations, for example, inpatient episodes by specialty or primary care appointments. Both primary and secondary outcomes will be stratified into predominantly physical or mental healthcare based on the primary reason for admission/attendance. The reason for healthcare use will also be further explored (eg, by body system/healthcare specialty).

## Statistical analyses
Analyses will be replicated across the three nations in each respective TRE.

### Objective 1
*Describe the background healthcare use in CYP before and during the pandemic*
Significant, dynamic changes in both healthcare access and healthcare-seeking behaviour have occurred across the course of the pandemic to date. As such, exploration of background healthcare use in CYP before and during the pandemic will help contextualise subsequent analyses. A descriptive, graphical analysis will be undertaken. Healthcare use (represented as cost) will be plotted for the period of 1 January 2019 to 1 May 2022 for all CYP. These data will be stratified by variables including age, sex, nation of residence, type of healthcare (primary or secondary care) and RT-PCR status (RT-PCR positive, RT-PCR negative and never tested). Reasons for healthcare visits will also be explored.

### Objective 2
*Compare healthcare use in CYP in the 12 months after testing positive, negative or not being tested for SARS-CoV-2 by RT-PCR to estimate the burden of healthcare use attributable to SARS-CoV-2.*
This analysis will focus on estimating the burden of CYP healthcare use which is attributable to SARS-CoV-2 infection in the 12 months after infection, whereas individual factors associated with healthcare use after infection will be explored in bjective 3. As well as total healthcare use in the 12 months following SARS-CoV-2 infection, we will also break this objective into 0–3, 3–6, 6–9 and 9–12 month brackets to examine how healthcare use changes across time.

A prospective matched cohort study will be undertaken. Matching will be undertaken for date of RT-PCR test, with iterative widening bands as necessary. This will account for availability of testing, access to healthcare, variation in incidence rates, emergence of viral variants, changes in SARS-CoV-2 treatment and systematically different characteristics in the tested population (compared with the untested population) as the pandemic progressed (online supplemental figure D). Ten RT-PCR test negative non-hospitalised control CYP will be matched without replacement for every RT-PCR positive case.

Stabilised inverse probability weights will be used to adjust for known confounder imbalance between cases and controls. The following variables will be explored: age, sex, SARS-CoV-2 vaccination status at the time of index RT-PCR test (considered vaccinated if ≥3 weeks since first dose), geographical region (health board/ Sustainability and Transformation Partnership (STP)) to account for regional differences in RT-PCR testing and availability of healthcare, previous healthcare contact (primary or secondary), chronic conditions, number of previous SARS-CoV-2 tests, socioeconomic status (quintiles of relevant national deprivation measure: Scottish Index of Multiple Deprivation (SIMD), Welsh Index of Multiple Deprivation and Lower layer Super Output Area) and Urban–Rural Index.

Factors are associated with being brought for RT-PCR testing (eg, public awareness and testing availability) may be different from those of exposure to SARS-CoV-2. A directed acyclic graph of factors associated with SARS-CoV-2 RT-PCR testing and healthcare use to consider in model building is shown in online supplemental figure D.

In contrast to adults, the median hospital length of stay due to SARS-CoV-2 in CYP is short, previously reported in the UK as 2 days (IQR 1–4).[22] As such, follow-up will start 14 days after testing positive for SARS-CoV-2 on RT-PCR which will enable us to look back and further stratify the exposure by SARS-CoV-2 severity (ie, community care, hospitalisation or critical care).

CYP in the control group may subsequently test positive for SARS-CoV-2 by RT-PCR. If this occurs during the RT-PCR testing period of interest (1 Januaty 2020–1 May 2021) they will become a case and follow-up commenced for 12 months (with appropriate matches for the date of the positive RT-PCR). If the control tests positive after 1 May 2021 (ie, after the RT-PCR testing period of interest), they will be censored and will not become a case. A graphical illustration of the potential CYP paths for this analysis is shown in figure 1.

As CYP who are brought for RT-PCR testing are systematically different to those who are not brought,[23] a sensitivity analysis will be undertaken to compare the RT-PCR positive cohort against the population of CYP who have never tested positive (ie, both RT-PCR negative and untested CYP), hereafter 'population controls'. RT-PCR positive CYP will be matched to ten population controls who were not hospitalised on the date of their matched case's RT-PCR.[7] Confounding will then be minimised as

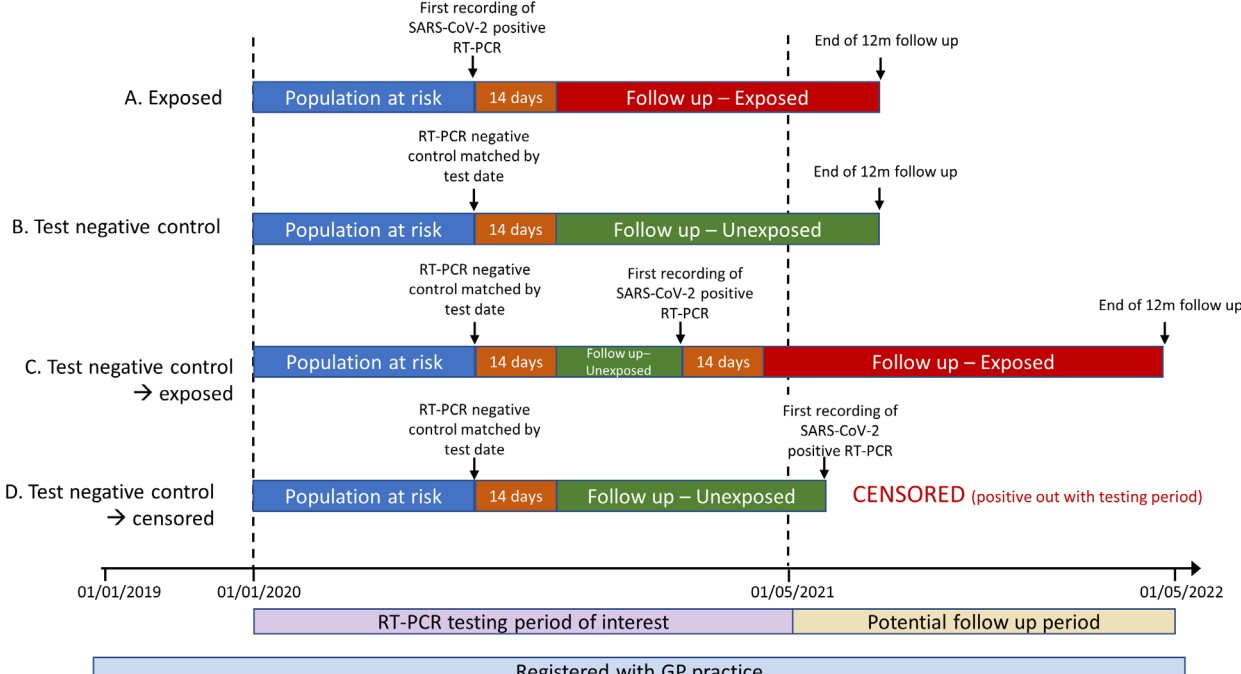

**Figure 1** Graphical illustrations of potential study scenarios with test negative controls. Example (A): Positive SARS-CoV-2 RT-PCR case. Individual (A) is followed up from 14 days after SARS-CoV-2 infection for 12 months. Examples (B–D): Test negative controls. Individual (B) is matched to an individual with SARS-CoV-2 infection and followed up from 14 days after matching for 12 months. Individual (C) is matched to an individual with SARS-CoV-2 infection and followed up from 14 days after matching until they are first recorded with SARS-CoV-2 infection themselves during the RT-PCR testing period of interest. At this point, they are censored from further follow-up as a test negative comparator and followed up as an exposed case from 14 days after infection for 12 months with appropriate matches for the date of positive RT-PCR. Individual (D) is matched to an individual with SARS-CoV-2 infection and followed up from 14 days after matching until they are first recorded with SARS-CoV-2 infection themselves. As this occurs after the RT-PCR testing period of interest, they are censored from further follow-up as an unexposed comparator. GP, general practitioner; RT-PCR, reverse transcription PCR.

described above. A graphical illustration of the potential CYP paths for this analysis is shown in online supplemental figure E.

The proportion of CYP with SARS-CoV-2 infection but without a positive RT-PCR (eg, tested by lateral flow (rapid antigen test) or untested asymptomatic cases) has increased across the pandemic.[3] As such, we will conduct quantitative bias analyses for unmeasured confounding using different estimates of undetected SARS-CoV-2 infection across the study period.

### Objective 3
*Identify factors associated with higher healthcare use (including having comorbidities) in the 12 months following SARS-CoV-2 infection*
Both regression and machine learning approaches will be undertaken to examine healthcare costs in the SARS-CoV-2 RT-PCR positive cohort. A multivariable regression model will be constructed with covariates including demographics (age, sex, socioeconomic status, Urban-Rural Index and health board/STP, pre-existing health status (chronic comorbidities, previous healthcare resource use, number of dispensed prescriptions, vaccination status and number of previous PCR tests), markers of severity of illness (community, hospital or intensive care within 14 days of index RT-PCR positive result) and PIMS-TS.

In order to examine CYP admitted due to SARS-CoV-2 (rather than those with incidental SARS-CoV-2 infection and another reason for admission), a sensitivity analysis will be performed excluding CYP with index SARS-CoV-2 RT-PCR undertaken 72 hours or less before an elective admissions, day case procedure or undertaken at any time during hospitalisation for trauma or emergency surgery.

We will then explore machine learning approaches to identify patterns of healthcare use over time following SARS-CoV-2 infection. Incorporating a machine learning analysis into this study will enable us to examine healthcare use across the 12-month period in a detailed way, investigating whether there are distinct groups of CYP who use healthcare in different ways over this period (ie, different trajectories). This might be in the level of healthcare used (eg, GP appointments, outpatient clinics or hospital admissions) or in when they use them (eg, one group may have higher 'upfront' healthcare use in the early period after SARS-CoV-2 infection while another has prolonged high healthcare throughout the 12-month period). Machine learning will allow us to cluster CYP into such trajectory groups and then explore whether particular characteristics are associated with each trajectory.

We will categorise CYP into groups based on their trajectories (ie, patterns of healthcare use). Both total

healthcare cost and types of healthcare (secondary care and scheduled primary care) will be considered. This will be done using three approaches: (1) latent growth mixture model of aggregated healthcare uses over a month,[24] (2) Bayesian categorical time series clustering of daily service uses of different types[25] and (3) centroid based clustering with dynamic time warping distance of smoothed healthcare use cost.[26] By modelling this time series of healthcare use, we will group patients into clusters with similar patterns, for example, one cluster may correspond to CYP who use general practices on a frequent basis but are not admitted to hospital while another cluster may belong to CYP who do not use general practices but attend outpatient clinics regularly.

After identifying CYP clusters, characteristics (including demographics, comorbidities and previous healthcare use) will be examined to identify any factors which may associated with higher healthcare needs post-SARS-CoV-2. These analyses will be stratified by hospitalisation (ie, hospital admission within 14 days of index RT-PCR positive result) or community care and by diagnosis of PIMS-TS. A sensitivity analysis excluding CYP with presumed incidental SARS-CoV-2 will be carried out as detailed above.

### Sensitivity analysis

It is likely that the majority of healthcare costs will be experienced within the first 3 months of SARS-CoV-2 infection.[9] Following on from objectives 2 and 3, we will extend the end date of the cohort to 3 months before the date of data extraction, and examine healthcare use in the 3 months following infection with SARS-CoV-2. This will enable us to examine healthcare with later Delta (B.1.617.2) and Omicron (B.1.1.529) variants.

### Anticipated limitations

While this protocol has been carefully developed there are anticipated limitations due to constraints of the data. Given the study period, it will also only be possible to examine the annual healthcare costs following infections with wild-type or Alpha (B.1.1.7) SARS-CoV-2 variant infections, which may not be the same as after Delta (B.1.617.2) or Omicron (B.1.1.529) variant infections. Data on SARS-CoV-2 viral variant is not consistently available for all CYP in this study. As such, time of RT-PCR testing will be used as a proxy for circulating viral variant at that time. The datasets included do not contain information on SARS-CoV-2 lateral flow testing results, which could result in misattribution of SARS-CoV-2 status in patients if RT-PCR testing was not performed. This is likely to particularly affect the later months of the study period where the highly transmissible Omicron variant was widespread and government advice no longer advocated RT-PCR following a positive lateral flow test in some situations.[27] In addition, the study will presume that healthcare services were available for anyone who needed them, but this may have been affected by local healthcare policies and patient health-seeking behaviour at different points during the pandemic. While this study will investigate healthcare use in the 12 months after SARS-CoV-2 infection, there will be other reasons for healthcare contacts in CYP which are not attributable to initial infection which cannot be accounted for in this analysis. This study only examines SARS-CoV-2 infections, not other viral or bacterial infections. It is possible that susceptibility to other infections is not the same in the SARS-CoV-2 RT-PCR positive and negative groups, potentially resulting in more healthcare contacts if one group has more non-SARS-CoV-2 infections over the study period than the other.

Finally, every observational study design has its own limitations. The design of this study relies on CYP registered with a GP which may introduce selection bias against those who are not registered (eg, in temporary accommodation) as well as the potential for recording biases in individuals coding the healthcare data.

### Patient and public involvement and engagement

This proposal was developed together with the Liverpool Generation-R Young Person's Advisory Group (YPAG), a group of engaged CYP aged between 12 and 21 years with lived experience of the SARS-CoV-2 pandemic. A member of the YPAG is also a coinvestigator and member of the steering committee, helping ensure the study is delivered appropriately and that decisions about study implementation are guided by meaningful patient and public involvement and engagement input. We will undertake two interactive workshops with the YPAG to cocreate educational materials for use in schools/science fairs. We will also use these workshops to discuss challenges regarding misinformation about SARS-CoV-2, strategies to correctly share information to young people using social media and the use of routine data in research. The YPAG have named the study—'Studying the Long-term Impact of COVID-19 in Kids (SLICK)' and chosen the logo (online supplemental figure F).

### ETHICS AND DISSEMINATION

This study was approved by the South Central—Oxford C—Health Research Authority Research Ethics Committee, approval reference number 13/SC/0149. This study involves routinely collected anonymised data and as such participant consent was not required.

The EAVE-II dataset was approved by the National Research Ethics Service Committee, South East Scotland 02 (REC number: 12/SS/0201) and the Public Benefit and Privacy Panel for Health and Social Care (reference number: 1920-0279).

EAVE-II was established to provide real-time surveillance and research on the SARS-CoV-2 pandemic in Scotland. The study includes the objective of understanding COVID-19 natural history and long-term sequelae through studying healthcare utilisation across the primary-secondary-tertiary care interface.

OpenSAFELY is a secure, transparent, open-source software platform for analysis of EHRs data with all activity publicly logged. The establishment of the OpenSAFELY platform was approved by the Health Research Authority (REC reference 20/LO/0651). The OpenSAFELY research platform adheres to the data protection principles of the UK Data Protection Act 2018 and the EU General Data Protection Regulation 2016 (for further details, please see online supplemental information).

The Welsh Con-COV research platform was created to determine demographic, socioeconomic and clinical risk factors for infection and mortality of COVID-19, to measure impact of COVID-19 on healthcare utilisation and long-term health, and to enable the evaluation of natural experiments of policy intervention.[28] The project (SAIL 0911) was approved by the independent Information Governance Review Panel. Investigation of the long-term healthcare burden of COVID-19 in children falls under this remit thus Con-COV is approved for use. Approved researchers are also able to access additional information within Con-COV that has been brought to SAIL under the Digital Economy Act to Accredited Researchers via the SAIL Databank.[29]

Guidelines for the Strengthening the Reporting of Observational Studies in Epidemiology (STROBE) and REporting of studies Conducted using Observational Routinely collected Data (via the COVID-19 extension) will be followed to report findings from this study. Findings will be presented at international conferences and published in peer-reviewed journals. Reports will also be prepared for policy makers. All analyses code will be made available through a public GitHub repository. In addition, a methods guide to producing harmonised metrics of paediatric healthcare costs across the three nations will be developed with associated code. Code lists to map and classify long term health conditions in paediatric populations in routine primary and secondary care datasets will be made available through OpenCodelists (www.opencodelists.org). Meta-data will be made available via the HDR-UK Innovation Gateway.

**Author affiliations**
[1]Centre for Medical Informatics, Usher Institute of Population Health Sciences and Informatics, The University of Edinburgh, Edinburgh, UK
[2]Department of Child Life and Health, The University of Edinburgh, Edinburgh, UK
[3]Usher Institute of Population Health Sciences and Informatics, The University of Edinburgh, Edinburgh, UK
[4]Department of Non-Communicable Disease Epidemiology, London School of Hygiene and Tropical Medicine, London, UK
[5]The DataLab, Nuffield Department of Primary Care Health Sciences, University of Oxford, Oxford, UK
[6]Centre for Population Health, Swansea University, Swansea, UK
[7]Department of Child Health, School of Medicine, Dentistry and Nursing, University of Glasgow, Glasgow, UK
[8]Institute of Cancer and Genetics, The University of Edinburgh, Edinburgh, UK
[9]School of Informatics, The University of Edinburgh, Edinburgh, UK
[10]Faculty of Humanities and Social Sciences, University of Liverpool, Liverpool, UK
[11]School of Medicine Dentistry and Nursing, University of Glasgow, Glasgow, UK
[12]Division of Genetics and Genomics, Roslin Institute, Edinburgh, UK
[13]MRC/CSO Social & Public Health Sciences Unit, University of Glasgow, Glasgow, UK
[14]Swansea University Medical School, Swansea University, Swansea, UK
[15]School of Health, Faculty of Health, Victoria University of Wellington, Wellington, New Zealand
[16]NIHR Health Protection Research Unit in Emerging and Zoonotic Infections, Liverpool, UK
[17]Respiratory Paediatrics, Alder Hey Children's Hospital, Liverpool, UK
[18]Health Data Research, Swansea University Medical School, Swansea, UK

**Acknowledgements** The cohorts included in this work have been funded separately as follows: ISARIC/CO-CIN is supported by grants from the National Institute for Health Research (award CO-CIN-01) and the Medical Research Council (grant MC_PC_19059) and by the National Institute for Health Research Health Protection Research Unit (NIHR HPRU) in Emerging and Zoonotic Infections at University of Liverpool in partnership with Public Health England (PHE), in collaboration with Liverpool School of Tropical Medicine and the University of Oxford (NIHR award 200907), Wellcome Trust and Department for International Development (215091/Z/18/Z), and the Bill and Melinda Gates Foundation (OPP1209135). EAVE II is funded by the Medical Research Council (MR/R008345/1) and supported by the Scottish Government. This work is supported by BREATHE—The Health Data Research Hub for Respiratory Health (MC_PC_19004). BREATHE is funded through the UK Research and Innovation Industrial Strategy Challenge Fund and delivered through Health Data Research UK. OpenSAFELY is jointly funded by UKRI (COV0076;MR/V015737/1) NIHR and Asthma UK-BLF and the Longitudinal Health and Wellbeing strand of the National Core Studies programme. The OpenSAFELY data science platform is funded by the Wellcome Trust. BG's work on better use of data in healthcare more broadly is currently funded in part by: the Wellcome Trust, NIHR Oxford Biomedical Research Centre, NIHR Applied Research Collaboration Oxford and Thames Valley, the Mohn-Westlake Foundation; all DataLab staff are supported by BG's grants on this work. SAIL Databank is funded by Health Care Research Wales and the analysis of this work was also funded by Health Care Research Wales through the Centre for Population Health and through Health Data Research Wales/N.Ireland, which receives its funding from HDR UK (HDR-9006) funded by the UK Medical Research Council, Engineering and Physical Sciences Research Council, Economic and Social Research Council, Department of Health and Social Care (England), Chief Scientist Office of the Scottish Government Health and Social Care Directorates, Health and Social Care Research and Development Division (Welsh Government), Public Health Agency (Northern Ireland), British Heart Foundation (BHF) and the Wellcome Trust. This work was supported by the Con-COV team funded by the Medical Research Council (grant number: MR/V028367/1).

**Contributors** OVS, NIL, EMH, PSH, MGS, SB, BG and ABD were responsible for conception of this project. OVS, LAT, AJW, MJS, JF, BG, SB and ABD will be responsible for data curation. OVS, NIL, EMH, LAT, AJW, MJS, JF, SS and ABD will be undertaking the analysis for this protocol. OVS, NIL, EMH, JKB, MGS, BG, SB, AS and ABD were responsible for securing funding for this project or its constituent cohorts. OVS, NIL, EMH, LAT, AJW, MJS, LP, JF, PSH, SS, JP, SJA, FFS, SVK, CRS, MGS, SB and ABD designed the analysis plan. OVS and ABD are providing administrative support to this project. LAT, AJW, MJS, JP, SJA, FFS, JKB, AA, RAL, MGS, BG, SB, AS and ABD are providing resources to this project. EMH, LAT, AJW, MJS, SS and BG are providing software for this project. MGS, AS and ABD are providing supervision. EMH, LAT, AJW, MJS, JF, TCW and SS will be responsible for data validation. OVS, EMH, AJW, MJS and SS are responsible for data visualisation. OVS, NIL, EMH, LAT, AJW, MJS, LP, PSH, SS and ABD wrote the original draft of this protocol and all authors were involved in the review and editing of this manuscript.

**Funding** This research is part of the Data and Connectivity National Core Study, led by Health Data Research UK in partnership with the Office for National Statistics and funded by UK Research and Innovation (grant ref MC_PC_20058). This work was also supported by The Alan Turing Institute via 'Towards Turing 2.0' EPSRC Grant Funding. The grant period spans 1 November 2021 to 30 September 2022.

**Competing interests** OVS reports an institutional payment from HDR-UK/Alan Turing for work on this study. LAT reports institutional contracts with UKRI, NIHR, MRC, institutional consulting fees from Bayer, support to attend MHRA meetings and unpaid membership of two non-industry funded trial advisory committees. MJS reports an institutional payment from HDR-UK/Alan Turing for work on this study. CRS reports institutional grants from MBIE, HRC and MRC. SVK reports funding from NRS, MRC and the Scottish Government Chief Scientist Office. He was co-chair of the Scottish Government's Expert Reference Group on Ethnicity

and COVID-19 and a member of the UK Scientific Advisory Group on Emergencies subgroup on ethnicity. MGS reports grants from NIHR, MRC and Health Protection Research Unit in Emerging & Zoonotic Infections, University of Liverpool. He also reports a role as Independent external and non-remunerated member of Pfizer's External Data Monitoring Committee for their mRNA vaccine program. He is Chair of Infectious Disease Scientific Advisory Board for Integrum Scientific LLC, Greensboro, NC, USA and director of MedEx Solutions Ltd. He reports minority stock ownership for Integrum Scientific LLC, Greensboro, NC, USA and majority stock ownership for MedEx Solutions Ltd. He also reports a gift from Chiesi Farmaceutici SPA to his institution of a clinical trial investigational medicinal product without encumbrance and distribution of same to trial sites. He is also a non-remunerated independent member of HMG UK Scientific Advisory Group for Emergencies (SAGE, COVID-19 Response) and HMG UK New Emerging Respiratory Virus Threats Advisory Group (NERVTAG). SB has received an institutional payment from HDR-UK/Alan Turing funding UOE Ref: 11563729 for work on this study. She also reports institutional payments from MRC, Welsh Government and NIHR. She is a member of the Population and Systems Medicine MRC board. AS reports an institutional payment from HDR-UK/Alan Turing and research grants for EAVE II and BREATHE Hub. He also reports non-remunerated positions on AstraZeneca's Thrombotic Thrombocytopenic Taskforce and Scottish and UK Government Advisory Committees. RAL is a member of the Welsh Government COVID-19 Technical Advisory Group. BG has received research funding from HDRUK, the Laura and John Arnold Foundation, the Wellcome Trust, the NIHR Oxford Biomedical Research Centre, the NHS National Institute for Health Research School of Primary Care Research, the Mohn-Westlake Foundation, the Good Thinking Foundation, the Health Foundation, and the World Health Organisation; he also receives personal income from speaking and writing for lay audiences on the misuse of science.

**Patient and public involvement** Patients and/or the public were involved in the design, or conduct, or reporting, or dissemination plans of this research. Refer to the Methods section for further details.

**Patient consent for publication** Not applicable.

**Provenance and peer review** Not commissioned; externally peer reviewed.

**ORCID iDs**
Olivia V Swann http://orcid.org/0000-0001-7386-2849
Ewen M Harrison http://orcid.org/0000-0002-5018-3066
Laurie A Tomlinson http://orcid.org/0000-0001-8848-9493
Alex J Walker http://orcid.org/0000-0003-4932-6135
Michael J Seaborne http://orcid.org/0000-0002-4921-7556
Srinivasa V Katikireddi http://orcid.org/0000-0001-6593-9092
Ashley Akbari http://orcid.org/0000-0003-0814-0801
Ronan Lyons http://orcid.org/0000-0001-5225-000X
Colin R Simpson http://orcid.org/0000-0002-5194-8083
Ben Goldacre http://orcid.org/0000-0002-5127-4728
Sinead Brophy http://orcid.org/0000-0001-7417-2858

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
