## [Reviewer comments · BMJ Open]

ARTICLE DETAILS

TITLE (PROVISIONAL)	Studying the Long-term Impact of Covid in Kids (SLICK). Healthcare use and costs in children and young people following community-acquired SARS-CoV-2 infection: protocol for an observational study using linked primary and secondary routinely collected healthcare data from England, Scotland and Wales.
AUTHORS	Swann, Olivia; Lone, Nazir; Harrison, Ewen; Tomlinson, Laurie; Walker, Alex; Seaborne, Michael; Pollock, Louisa; Farrell, James; Hall, Peter; Seth, Sohan; Williams, Thomas; Preston, Jenny; Ainsworth, Jennifer; Semple, Freya; Baillie, J; Katikireddi, Srinivasa; Akbari, Ashley; Lyons, Ronan; Simpson, Colin; Semple, Malcolm G; Goldacre, Ben; Brophy, Sinead; Sheikh, Aziz; Docherty, Annemarie

VERSION 1 – REVIEW

REVIEWER	Chris Rees Boston Children s Hospital, Pediatric Emergency Medicine
REVIEW RETURNED	06-May-2022

GENERAL COMMENTS	The authors report a study protocol to determine healthcare use among children with SARS-CoV-2 in three nations. This is a very well written protocol with important potential implications. I applaud the authors for undertaking this important, large-scale work. My comments below are fairly minor, but I wonder if the investigators have access to patient vaccination status against SARS-CoV-2? It may be an important subanalysis to do to determine differences in the outcomes of interest among children who were vaccinated and later tested positive for SARS-CoV-2. Abstract: -What is the rationale for including SARS-CoV-2 test data from 01/01/2019? If SARS-CoV-2 was first identified in December 2019, isn't there a long period without any potential positive tests leading up to that time? Perhaps the authors propose collecting data other than SARS-CoV-2 during that time, but the first sentence of the Methods and Analysis section is unclear in this regard in the Abstract (though it is clearly described in the Methods). Strengths and Limitations: -Perhaps "lateral flow testing" is commonly used in some parts of the world, but I am not familiar with this phrase. Consider defining or describing this. Introduction: -Clearly written and concisely makes the case for the proposed work.
---

	Methods: -It is unclear why under "Objective 1-Objective-specific inclusion criteria" the second and third bullet points are different. Please clarify. -I think a deeper description of the Trusted Research Environments: OpenSAFELY in England, Secure Anonymized Information Linkage in Wales, and the EAVE-Il platforms is warranted. These seem to be administrative databases, but perhaps some description of how these databases are constructed and for what purpose would help the reader understand these data sources. -Under "Statistical Analyses", line 5: Please define TRE. I am not sure it was defined previously in the paper and is not a common acronym.
--	---

REVIEWER	N Toepfner Carl Gustav Carus University Hospital
REVIEW RETURNED	05-Aug-2022

GENERAL COMMENTS	It is an interesting study protocol and an important approach towards the long-term impact of COVID in children. However, there are a couple of aspects regarding the planned study which are not addressed by the study protocol: A major point seems the 12 months interval following SARS-CoV-2 infection. As 12 months is a long period in which the children can acquire a lot of other infections. How can the authors attribute all effects in this interval to SARS-CoV-2 (e. g. p5, l41)? Will an adjustment e. g. be carried out for other viral infection in comparison between case and control group? Please explain in more details which and how adjustments are planned. Even with adjustments it cannot be set as certain, that the susceptibility for other viral infections is the same in the SARS-CoV-2 positive and negative group. Therefore please also discuss these aspects in the study limitations. Additionally, at a single timepoint of "after 12 months" just a very special and limited phenomenon of healthcare use and costs will be examined. The intensity of healthcare use and costs will potentially vary over time. Are interim analysis planned? If yes, please specify or explain, why they are not planned. p5, l 33 Please add, if the RT-PCR was variant specific and include this information in the discussion of potential study limitations. p7, l 16 and e.g. page 8, l 56 etc. The strength of this study seems exaggerated. It does not generally lead to a reduction in selection and response biases, but might just add another perspective with its own selection and recording biases etc. These need to be discussed in many more details. Please cite more up to date research for relevant statements and assumptions e.g. more than one older review on p8 l 41 for this continuously evolving field of research. p 9 l 8 How will machine learning be used - please specify in more detail. p10 l 15 is there a typing error? Or is 01/01/21 the correct time frame? Could the authors please explain how they plan to deal with vaccination effects in more detail? Will different subgroups of age be analyzed? Please specify. Please add more details on the research ethics (e.g. participant consent, ethics approval)
--

VERSION 1 – AUTHOR RESPONSE

Reviewer: 1

Dr. Chris Rees, Boston Children's Hospital

Comments to the Author:

Point 3. The authors report a study protocol to determine healthcare use among children with SARS-CoV-2 in three nations. This is a very well written protocol with important potential implications. I applaud the authors for undertaking this important, large-scale work.

Thank you for your kind comments.

Point 4. My comments below are fairly minor, but I wonder if the investigators have access to patient vaccination status against SARS-CoV-2? It may be an important subanalysis to do to determine differences in the outcomes of interest among children who were vaccinated and later tested positive for SARS-CoV-2.

Thank you for raising this important point which we have considered in this study protocol. SARS-CoV-2 vaccination status is included as a variable in Table 1 (Groupings of variables by source). Vaccination status is subsequently described in the methods for Objective 2 as one of the variables to be adjusted for after matching on the date of RT-PCR test: " ... *will include the following: age, sex, SARS-CoV-2 vaccination status at the time of index RT-PCR test (considered vaccinated if ≥ 3 weeks since first dose)* ... "

Vaccination status is also included as a variable in Objective 3 : " *A multivariable regression model will be constructed with covariates including demographics (age, sex, socioeconomic status, urban-rural Index and health board / STP, pre-existing health status (chronic comorbidities, previous health care resource use, number of dispensed prescriptions, vaccination status* ... " We hope that Dr Rees finds this acceptable.

Point 5. Abstract: What is the rationale for including SARS-CoV-2 test data from 01/01/2019? If SARS-CoV-2 was first identified in December 2019, isn't there a long period without any potential positive tests leading up to that time? Perhaps the authors propose collecting data other than SARS-CoV-2 during that time, but the first sentence of the Methods and Analysis section is unclear in this regard in the Abstract (though it is clearly described in the Methods).

Thank you for flagging this and we apologise for the confusion. We chose to start the study period from 01/01/19 to ensure that we had at least a year of data on healthcare use before the pandemic began for each child or young person (CYP). This approach is outlined in the methods section under the Study Period subheading: " *Inclusion of the time frame 01/01/19 to 01/01/20 will provide at least a*

year of data on pre-pandemic data on healthcare use for each CYP.” Please also note that this sentence originally read “01/01/19 to 01/01/21” which was a typo as picked up by Dr Toepfner, see Point 20 , and has now been corrected - our apologies.

Inclusion of data from a year before the SARS-CoV-2 pandemic will allow us to consider pre-pandemic healthcare use as a variable in identifying CYP with high healthcare use post SARS-CoV-2 infection as explored in Objective 3. This has been outlined in the first paragraph of Objective 3 methods “*A multivariable regression model will be constructed with covariates including (...) pre-existing health status (chronic comorbidities, previous health care resource use ...*”

To improve clarity, we have added a sentence to the abstract which now reads: “*We will use anonymised individual-level, population-scale national data linking demographics, comorbidities, primary and secondary care use and mortality between 01/01/2019-01/05/2022. SARS-CoV-2 test data will be linked from 01/01/20-01/05/2022.*” We hope this has made our intentions clearer.

Point 6. Strengths and Limitations: Perhaps “lateral flow testing” is commonly used in some parts of the world, but I am not familiar with this phrase. Consider defining or describing this.

Please accept our apologies for any confusion here. We have added “*rapid antigen testing*” in brackets after the phrase “*lateral flow testing*” where it first appears in the text. Please let us know if an alternative phrase would be preferable.

Point 7. Introduction: Clearly written and concisely makes the case for the proposed work.

Thank you.

Point 8. Methods: It is unclear why under “Objective 1-Objective-specific inclusion criteria” the second and third bullet points are different. Please clarify.

Thank you for picking this up. These are pandemic years one and two and we have now added in brackets after each date range “Pandemic Year 1” and “Pandemic Year 2” to clarify this for the reader.

Point 9. I think a deeper description of the Trusted Research Environments: OpenSAFELY in England, Secure Anonymized Information Linkage in Wales, and the EAVE-II platforms is warranted. These seem to be administrative databases, but perhaps some description of how these databases are constructed and for what purpose would help the reader understand these data sources. -Under “Statistical Analyses”, line 5: Please define TRE. I am not sure it was defined previously in the paper and is not a common acronym.

Trusted Research Environments are integral to this paper, thank you for the opportunity to expand this section. The acronym TRE is defined earlier in the manuscript under the Data Sources and Validation hearing: *“Data will be held securely and analyses conducted within nation-specific Trusted Research Environments (TREs) ... “*

To help orientate the reader, we have added an additional line to this section to outline the role of TREs: *“TREs provide secure computing environments which hold data remotely and enable access for analysis without the data itself ever leaving the secure site.”*

We have also added some brief background information for each of the there TREs in this study to help provide context for the data:

“OpenSAFELY is a secure, transparent, open-source software platform for analysis of electronic health records data allowing detailed analysis of pseudonymised primary care patient records in England. Other datasets are linked within the same environment using a matching pseudonym derived from the National Health Service (NHS) number.

SAIL Databank brings together electronically-held, person-based, routinely-collected demographic and clinical data across Wales for the purpose of conducting and supporting health-related research, which are pseudonymised using Anonymous Linking Fields.

EAVE-II is a national linked dataset of patient-level primary care data, out-of-hours, hospitalisation, mortality and laboratory data across Scotland. Data is held securely and analysed in the Public Health Scotland TRE.”

Reviewer: 2

Dr. N Toepfner, Carl Gustav Carus University Hospital

Comments to the Author:

Point 10. It is an interesting study protocol and an important approach towards the long-term impact of COVID in children. However, there are a couple of aspects regarding the planned study which are not addressed by the study protocol:

Point 11. A major point seems the 12 months interval following SARS-CoV-2 infection. As 12 months is a long period in which the children can acquire a lot of other infections. How can the authors attribute all effects in this interval to SARS-CoV-2 (e. g. p5, l41)?

Thank you for this comment with which we agree. Whilst this is an observational study, the use of matching for time of RT-PCR testing, inverse probability weighting for confounding variables and subsequent analyses are being undertaken to try and ascertain whether SARS-CoV-2 infection is independently associated with healthcare utilisation. However, we fully agree with Dr Toepfner that we cannot attribute all effects in the subsequent 12 month period to SARS-CoV-2 infection.

To clarify the limitations of our approach, we have added a sentence to the Anticipated Limitations section: *“Whilst this study will investigate healthcare use in the 12 months after SARS-CoV-2 infection, there will be other reasons for healthcare contacts in CYP which are not attributable to initial infection which cannot be accounted for in this analysis.”*

In response to this comment, we will also overtly explore healthcare use over shorter time periods from infection. We have added a line to Objective 2 methods section to highlight this:

“As well as total healthcare use in the 12 months following SARS-CoV-2 infection, we will also break this objective into 0-3, 3-6, 6-9 and 9-12 month brackets to examine how healthcare use changes across time.”

The trajectories of healthcare use are also being explored using machine learning which will examine how healthcare is used over the 12 months following SARS-CoV-2 infection in more detail. This will allow us to investigate risk factors for healthcare use at different time points over the year (e.g., are some demographic or comorbid variables associated with initial high healthcare use, while others are associated with prolonged high healthcare use across the year?). The machine learning approach is discussed in more detail in Point 19 below.

Point 12. Will an adjustment e. g. be carried out for other viral infection in comparison between case and control group?

This is an excellent point and one which was discussed at length by the study group. After careful thought it was decided that inclusion of an infected control group would obfuscate rather than improve the analysis. This decision was based on the dual issues of infection dynamics of viruses other than SARS-CoV-2 changing significantly during the pandemic period (making comparison with another virus extremely challenging) as well as changes in the availability of healthcare during the pandemic (meaning that the healthcare use of CYP with historical viral infection controls would not be comparable to that during the pandemic). As such, we elected not to use a control group with an alternate infection. In addition, as we are matching groups by date of RT-PCR testing, we hope this should account for circulating seasonal viruses to some extent.

Point 13. Please explain in more details which and how adjustments are planned.

Thank you for the opportunity to make this section clearer. CYP are matched on date of RT-PCR test which is a composite proxy variable for SARS-CoV-2 community prevalence, viral variant, testing availability, healthcare availability, non-pharmacological interventions, COVID-specific treatment availability and public awareness (as detailed in *Supplementary Figure D – Directed acyclic graph factors associated with SARS-CoV-2 RT-PCR testing and healthcare use*). An additional figure call out has been added to the Objective 2 methods section to improve clarity here.

After matching for time, we will undertake adjustment to balance potential confounders and improve comparability of the RT-PCR positive and negative groups. Since submitting the protocol, we have considered different approaches for this and elected to use inverse probability weighting. The methods section of Objective 2 has been updated to reflect this (changes in bold) and the method explained in more detail:

“Stabilised inverse probability weights will be used to adjust for known confounder imbalance between cases and controls. The following variables will be explored: age, sex, SARS-CoV-2 vaccination status at the time of index RT-PCR test (considered vaccinated if ≥3 weeks since first dose), geographical region (health board / Sustainability and Transformation Partnership (STP)) to account for regional differences in RT-PCR testing and availability of healthcare, previous healthcare contact (primary or secondary), chronic conditions, number of previous SARS-CoV-2 tests, socioeconomic status (quintiles of relevant national deprivation measure: Scottish Index of Multiple Deprivation (SIMD), Welsh Index of Multiple Deprivation (WIMD) and Lower layer Super Output Area (LSOA)) and urban-rural index. ”

Point 14. Even with adjustments it cannot be set as certain, that the susceptibility for other viral infections is the same in the SARS-CoV-2 positive and negative group. Therefore please also discuss these aspects in the study limitations.

This is an important point and we agree with Dr Toepfner. We have added the following to the Anticipated Limitations section:

“This study only examines SARS-CoV-2 infections, not other viral or bacterial infections. It is possible that susceptibility to other infections is not the same in the SARS-CoV-2 RT-PCR positive and negative groups, potentially resulting in more healthcare contacts if one group has more non-SARS-CoV-2 infections over the study period than the other.”

Point 15. Additionally, at a single timepoint of "after 12 months" just a very special and limited phenomenon of healthcare use and costs will be examined. The intensity of healthcare use and costs will potentially vary over time. Are interim analysis planned? If yes, please specify or explain, why they are not planned.

Please see responses to Point 11 above and Point 19 below which discuss the use of healthcare varying over time.

Point 16. p5, l 33 Please add, if the RT-PCR was variant specific and include this information in the discussion of potential study limitations.

Unfortunately viral variant data were not consistently available for all CYP in this study. Instead, we will use time of RT-PCR as a proxy for viral variant (*Supplementary Figure D*). A sentence has been added to the limitations section to this effect:

“Data on SARS-CoV-2 viral variant is not consistently available for all CYP in this study. As such, time of RT-PCR testing will be used as a proxy for circulating viral variant at that time.”

Point 17. p7, l 16 and e.g. page 8, l 56 etc. The strength of this study seems exaggerated. It does not generally lead to a reduction in selection and response biases, but might just add another perspective with its own selection and recording biases etc. These need to be discussed in many more details.

We fully appreciate this comment and agree that there is no perfect observational study. However, we do feel that this approach should reduce some of the inherent biases present in previous studies of reported symptoms which have often used voluntary questionnaires.

We have made the following changes:

- a) Under Anticipated Limitations, we have removed “to reduce bias” from the sentence *“Whilst this protocol has been carefully developed there are anticipated limitations due to constraints of the data.”*
- b) We have removed the sentence *“Reduction in selection and response biases present in much of the existing literature examining persistent symptoms post SARS-CoV-2 infection in CYP.”* from the Article Summary section.

- c) We have amended the sentence in the introduction from *“Using routinely collected anonymised electronic health record (EHR) data at an individual-level, population-scale matched by SARS-CoV-2 RT-PCR status to examine healthcare use after SARS-CoV-2 infection in CYP will significantly reduce many of the biases seen in studies to date”* to *“Using routinely collected anonymised electronic health record (EHR) data at an individual-level, population-scale matched by SARS-CoV-2 RT-PCR status to examine healthcare use after SARS-CoV-2 infection in CYP offers an alternative method to questionnaire or clinic-based symptom reporting after SARS-CoV-2.”* We also have added a sentence to the Anticipated Limitations section: *“Finally, every observational study design has its own limitations. The design of this study relies on CYP registered with a GP which may introduce selection bias against those who are not registered (e.g. in temporary accommodation) as well as the potential for recording biases in individuals coding the healthcare data.”*

We hope Dr Toepfner feels that these changes better reflect the study description.

Point 18. Please cite more up to date research for relevant statements and assumptions e.g. more than one older review on p8 | 41 for this continuously evolving field of research.

This work was submitted for review in March 2022 at which point there was little information on this area. Thank you for the chance to update the background with recent relevant studies which we have included in the introduction:

“Data on long term healthcare use following SARS-CoV-2 infection is beginning to emerge, although most studies have focused on adults rather than CYP. One large study of American adults (n= 5,064,270) reported an increase in outpatient healthcare use in the six months following SARS-CoV-2 infection (hazard ratio of 1.20 (1.19–1.21))⁷. Another American study (n=250,514) found COVID-19 diagnosis was associated with an additional 0.7269 (95% CI, 0.7088 to 0.7449) monthly healthcare visits (combined inpatient and outpatient visits excluding respiratory healthcare contacts) in the six months after diagnosis⁸. This study did include some CYP (n not given) and reported that healthcare use post-COVID-19 diagnosis increased slightly from two to five months after diagnosis for those ≤17 years old, but returned to pre-diagnosis baseline levels by six months.

One Norwegian study examined healthcare use in CYP aged 1-19 years (n=706,885) for six months from SARS-CoV-2 testing and reported an increase in primary healthcare use for all ages during the first one to four weeks following a positive test compared with CYP who tested negative⁹. These presentations were predominantly respiratory. This increase in healthcare use was more sustained in younger CYP, while those aged 16-19 years returned to baseline healthcare use by five to eight weeks. The study did not find any increase in use of specialist care for any age group.”

Point 19. p 9 | 8 How will machine learning be used - please specify in more detail.

We are excited by the incorporation of machine learning methods to examine data in ways which would be hard to disentangle using more classical analyses. We have added the following paragraph to the machine learning methods paragraphs to improve clarity for the reader and explain how they add to the existing analyses:

“Incorporating a machine learning analysis into this study will enable us to examine healthcare use across the 12 month period in a detailed way, investigating whether there are distinct groups of CYP who use healthcare in different ways over this period (i.e. different trajectories). This might be in the level of healthcare used (e.g. GP appointments, outpatient clinics or hospital admissions) or in when they use them (e.g. one group may have higher “upfront” healthcare use in the early period after SARS-CoV-2 infection while another has prolonged high healthcare throughout the 12 month period). Machine learning will allow us to cluster CYP into such trajectory groups and then explore whether particular characteristics are associated with each trajectory “

Point 20. p10 l 15 is there a typing error? Or is 01/01/21 the correct time frame?

Thank you for spotting this – this was indeed an error and has now been corrected to 01/01/20

Point 21. Could the authors please explain how they plan to deal with vaccination effects in more detail? Will different subgroups of age be analyzed? Please specify.

Thank you for this important point and reminding us that the comorbidity and age groups of CYP eligible for vaccination changed with time over the pandemic. Please see response to Dr Rees (Reviewer 1)’s comment above. Vaccination status will be adjusted for as part of the inverse probability weighting (IPW) methods to balance confounding factors between groups. As such, the effect of vaccination will be adjusted for in the final analysis without an age-stratified subanalysis.

Point 22. Please add more details on the research ethics (e.g. participant consent, ethics approval) A brief ethical statement is found at the end of the abstract with full details in the “Ethics and Dissemination” section. We have added more information for EAVE-II in this section:

“EAVE-II was established to provide real time surveillance and research on the SARS-CoV-2 pandemic in Scotland. The study includes the objective of understanding COVID-19 natural history and long term sequelae through studying healthcare utilisation across the primary-secondary-tertiary care interface.”

We have also added a long form version of the OpenSAFELY information governance and ethical approval to the Supplementary Information for more detailed information.

As the SLICK study is based on routinely collected and anonymised data, no participant consent is necessary. We have added a statement to this effect to the Ethics and Dissemination section for clarity: *“This study involves routinely collected anonymised data and as such participant consent was not required”*

VERSION 2 – REVIEW

REVIEWER	Chris Rees Boston Children s Hospital, Pediatric Emergency Medicine
REVIEW RETURNED	21-Sep-2022

GENERAL COMMENTS	The authors have been very responsive to my original comments. I have no further suggestions.
---

REVIEWER	N Toepfner Carl Gustav Carus University Hospital
REVIEW RETURNED	04-Oct-2022

GENERAL COMMENTS	Revision has significantly improved the manuscript. Well done!
--